# Quantitative Assessment of Major Biosecurity Challenges of Poultry Production in Central Ethiopia

**DOI:** 10.3390/ani13233719

**Published:** 2023-11-30

**Authors:** Hika Waktole, Tsedal Muluneh, Yemsrach Miressa, Sisay Ayane, Gebreyohannes Berhane, Tadele Kabeta, Bizunesh Mideksa Borena, Kebede Amenu, Hagos Ashenafi, Gunther Antonissen

**Affiliations:** 1Department of Microbiology, Immunology and Veterinary Public Health, College of Veterinary Medicine and Agriculture, Addis Ababa University, Bishoftu P.O. Box 34, Ethiopia; kebede.amenu@aau.edu.et; 2Department of Pathobiology, Pharmacology and Zoological Medicine, Faculty of Veterinary Medicine, Ghent University, 9820 Merelbeke, Belgium; gunther.antonissen@ugent.be; 3Department of Animal Production Studies, College of Veterinary Medicine and Agriculture, Addis Ababa University, Bishoftu P.O. Box 34, Ethiopia; tsedalmuluneh19@gmail.com (T.M.); gebreyohannes.berhane@aau.edu.et (G.B.); 4School of Veterinary Medicine, College of Agriculture and Veterinary Medicine, Jimma University, Jimma P.O. Box 378, Ethiopia; yemsrachmiressa@gmail.com (Y.M.); firanfiri.04@gmail.com (T.K.); 5Department of Veterinary Sciences, School of Veterinary Medicine, Ambo University, Ambo P.O. Box 19, Ethiopia; sisayayane12@gmail.com (S.A.); bizunesh.mideksa@ambou.edu.et (B.M.B.); 6International Livestock Research Institute (ILRI), Addis Ababa P.O. Box 5689, Ethiopia; 7Aklilu Lemma Institute of Pathobiology, Addis Ababa University, Addis Ababa P.O. Box 1176, Ethiopia; hagos.ashenafi1@aau.edu.et

**Keywords:** biosecurity, central Ethiopia, external biosecurity, internal biosecurity, poultry farms

## Abstract

**Simple Summary:**

Evaluation of biosecurity measures remains essential for successful control and prevention of diseases in a country like Ethiopia, where the poultry sector is flourishing. This research was conducted from October 2020 to June 2022 and aimed to assess the level of implementation of biosecurity practices of 226 poultry farms belonging to different scales found in three selected areas of central Ethiopia, including Addis Ababa, Bishoftu and West of Shaggar City. The study showed a very low overall biosecurity practices score compared to the global average. All three study areas had rather similar levels of biosecurity practices, being statistically insignificant. The majority of the external biosecurity components were highly disrupted and far below the global average. Similarly, the internal biosecurity practices were found to be still lower than the global average. Thus, weak and relaxed biosecurity practices were evident among poultry farms in central Ethiopia. Understanding the poor status of the implementation of appropriate biosecurity practices will help to design and develop strategies that safeguard the poultry production sector from the impact of various deadly diseases. The findings emphasize the need for active involvement of all stakeholders in the poultry production value chain towards boosting the productivity of the sector in Ethiopia.

**Abstract:**

The present study aims to assess the level of implementation of biosecurity practices of small-, medium- and large-scale poultry farms in central Ethiopia. A cross-sectional study design was implemented involving 226 poultry farms (153 small-, 53 medium- and 20 large-scale farms) in three selected areas of central Ethiopia, including Addis Ababa, Bishoftu and West of Shaggar. The results revealed a very low overall biosecurity score of 43.1% for central Ethiopia compared to the global average score of 64.3%. No significant difference (*p* > 0.05) in the overall biosecurity practices among the three study sites (36.1% for Addis Ababa, 49% for Bishoftu and 34.9% for West of Shaggar) was observed. Six of the eight external biosecurity components with an overall score of 40.7% as compared to the global average of 64.6% were highly disrupted and scored lower values. With regard to internal biosecurity disease management, cleaning and disinfection practices were found to be 52.6% as opposed to the global average of 64%. The poor biosecurity level among poultry farms of central Ethiopia clearly demonstrates the urgent need for the implementation of appropriate biosecurity practices through the active involvement of all stakeholders to combat the impact of various diseases and boost the productivity of the sector.

## 1. Introduction

With the growing human population and standard of living, the consumption of chicken meat and table eggs is becoming increasingly important in Ethiopia [1,2]. Chicken production is one of the promising sectors for improving the food and nutrition security of the country, mainly due to the fast growth, low feed conversion ratios and relatively small land requirements. Chicken meat is a relatively cheap and affordable source of protein compared to other animal products, such as beef [3]. In Ethiopia, the poultry sector is rapidly growing and predominantly characterized by backyard or village production systems and only a small proportion of small- to large-scale commercial production systems [4]. The chicken population of the country is estimated at about 57 million, with 78.9% indigenous, 12% exotic and 9.1% hybrid breeds [5]. However, the poultry sector in Ethiopia is facing various challenges, including a high prevalence of diseases and management-related problems [6,7]. Among the major contributing factors to the widespread occurrence of diseases is the lack of proper understanding and implementation of biosecurity practices. Improved biosecurity could significantly reduce diseases and managemental problems and boost farm productivity [8,9]. Moreover, there is also a lack of a policy framework and guidelines for addressing poultry farm biosecurity practices.

Good biosecurity has great potential in terms of reducing, controlling and preventing the spread of pathogens within and between poultry flocks, but also the transmission of zoonotic diseases [10,11]. The advantages of proper poultry farm biosecurity are associated with the reduction in costs associated with diseases and subsequently increasing profits, leading the poultry industry to thrive. Biosecurity involves a comprehensive range of procedures to limit the introduction of infection into poultry production units by means of three major components: segregation, to keep contaminated people, animals and materials away from uninfected birds; cleaning, to remove most of the contaminated organic matter; and disinfection, which, if properly implemented, destroys the pathogens [12,13]. It is clearly demonstrated that biosecurity measures at the level of the farm are the foundation of maintaining the health of birds and also the safety of the animal-derived products along the entire value chain of poultry production [14,15].

In Ethiopia, similar to the case in Sub-Saharan African countries, few studies conducted have revealed that biosecurity measures are not properly implemented [16,17]. Reliable and exhaustive information is scarce on the biosecurity status of commercial poultry farms, which are mainly concentrated in central Ethiopia. Systematic investigation into the biosecurity status of commercial farms is crucial to devise preventive measures taking into account the farms’ scales [3,18,19,20]. The main objective of the present study was to appraise and explore the status of biosecurity practices among poultry farms of selected areas of central Ethiopia.

## 2. Materials and Methods

### 2.1. Study Area

The study was carried out in three selected areas of central Ethiopia, mainly in Addis Ababa and its surroundings, extending up to 100 km radius, more specifically in Addis Ababa, Bishoftu and West of Shaggar (Holeta and Ambo) and Bishoftu (Figure 1).

Commercial poultry production is very common in central Ethiopia. Majority of the small-, medium- and large-scale poultry farms found in the study area were considered. The main selection criteria for the inclusion of poultry farms were based on snowball sampling procedure considering accessibility of farms and willingness of farmers to participate in the study.

### 2.2. Study Design and Study Population

Assessment of biosecurity practices on small-, medium- and large-scale commercial chicken farms was carried out in the period October 2020 until June 2022 following a cross-sectional study design. A total of 226 farms (153 small-, 53 medium- and 20 large-scale farms) were studied from Addis Ababa (56 farms), Bishoftu (70 farms) and West of Shaggar (100 farms), representing central Ethiopia. The total poultry population of Ethiopia was estimated to be 57 million chickens in the year 2021/22 [5]. The majority of commercial poultry farms are concentrated in urban and peri-urban areas of central Ethiopia [21]. The flock sizes were categorized as small-scale, consisting of 100–1000 birds, medium-scale, with 1001–10,000 birds, and large-scale, above 10,001 birds [22]. The study considered a total of 157, 37 and 32 poultry farms kept for the purposes of egg, meat and mixed production, respectively. Prior to the biosecurity evaluations, informal visits were made to the poultry farms to create trusting relationships with the farm owners and/or workers, to explain the objectives of the study and the commitment towards confidentiality. Afterward, the biosecurity audit was performed during an unannounced farm visit.

### 2.3. Quantification of Biosecurity Score

The poultry farm biosecurity parameters and practices were evaluated by a scientific risk-based scoring system based on the Biocheck.UGent™ tool (Merelbeke, Ghent University, Belgium) (https://biocheck.ugent.be/en (accessed on 27 November 2023)). This tool to assess biosecurity has been validated for poultry, pigs and cattle [15,23,24,25]. However, questions were modified with relevance to the poultry production system in Ethiopia and along with the specific study sites. Questionnaires comprised both external and internal biosecurity parameters, with subsections of the questionnaire including general characteristics of farms and respondents’ profile, chicken farm characteristics, external biosecurity (chicken purchasing practices, feeding and watering management, waste and manure management practice, farm entry restrictions, inter-farm material sharing practice, farm infrastructure status, control of biological vectors and farm relative location from the nearby farm and distance from the main road) and internal biosecurity (disease management and cleaning and disinfection measures). The level of biosecurity practices was quantified by converting the answers to 55 questions into a score between 0 (=total absence of biosecurity measures) and 100 (=full presence of biosecurity measures) [15]. Attributed to the variation in the relative weight, the external biosecurity score accounted for 80% and that of the internal score 20% towards the quantification of the total biosecurity score.

For ease of interpretation of the results, category and subcategory scores were recalculated each time to a score of 100 and presented as a percentage [15]. The final score was obtained by adding the scores of the eight external and two internal biosecurity subcategories. For external biosecurity, the following subcategories were included: (1) chicken purchasing practice, (2) feed and water management, (3) waste management, (4) restriction to farm entry, (5) inter-farm material sharing, (6) status of farm infrastructure, (7) control of biological vectors and (8) relative location of farm according to the recommended guidelines. The internal biosecurity practices were assessed related to disease management and farm cleaning practices [15]. The definitions of each of the external and internal subcategories are shown in Table 1. The responses under the subcategories were computed to represent the component average. After completion of the biosecurity quantification, a score becomes available for external, internal and overall biosecurity on the poultry farms in each study site. Finally, the computed poultry biosecurity score was compared to that of the global average [24].

### 2.4. Data Collection and Analysis

The organized questionnaire was piloted prior to assessing the actual study on 15 non-participating farms to improve the clarity of questions, their relevance and to assess the efficient use of time. Additionally, the date and the time of the farm visit, the location of the farm, general characteristics of the farm and respondents were all recorded. The finalized form for each farm was uploaded to the project created on the KoBoToolbox server (open-source tool). The raw data from the questionnaire survey were obtained as Excel files from the Kobo Toolbox server to be coded, processed and imported into data analysis tools. Map of the study area was generated with QGIS version 3.14 (QGIS Development Team, 2009. QGIS Geographic Information System. Open Source Geospatial Foundation. http://qgis.org). The level of differences among study sites, type of production and farm sizes for external and internal biosecurity scores and the associated 10 components, and the characteristics of the studied chicken farms were computed using chi-squared tests and *t*-tests by using STATA version 17 (Stata, College Station, TX, USA).

## 3. Results and Discussion

### 3.1. Characteristics of the Study Participants and Poultry Farms

The major demographic characteristics of the participants are presented in Table 2. It was noted that many of the respondents were farm owners, male, between 26 and 35 years old and their educational status lies between primary and secondary school.

### 3.2. The Characteristics of the Studied Poultry Farms

The characteristics of the poultry farms and chickens in the three selected sites of central Ethiopia were analyzed and the findings depicted in Table 3.

The results revealed that the purpose of chicken farming was predominately for egg production (157 out of 226). Out of 226 poultry farms, the flock size consisted of 153 small-, 53 medium- and 20 large-scale farms. Most of the chicken production in the country is small-scale, and it is only recently that commercial poultry farms are becoming very common, especially in the central highlands of Ethiopia and around major cities [21,26]. In the present study, Bovans Brown was the dominant breed, followed by Cobb 500, Sasso and Lohmann Brown, in the farms of central Ethiopia, as illustrated in Table 3. In Ethiopia, it is a common practice to rear different purposes of chickens in various scales of commercial farms. However, in the conventional or village production system, the indigenous breeds are quite frequently used [21,27]. The housing of studied farms was predominately deep litter system (χ^2^ = 19.91; *p* < 0.001) in 47%, 55% and 99% of the poultry farms in Addis Ababa, Bishoftu and West of Shaggar, respectively. It was disclosed that deep litter housing systems possess the drawback of permitting the accumulation of noxious gases, such as ammonia, pathogenic bacteria and larvae of parasites, with a potential negative influence on biosecurity practices [28,29].

Interestingly, some of the poultry farmers in the study areas kept different breeds of chickens on the same farm. Moreover, the majority of the farms (100% in Addis Ababa, 41.4% at Bishoftu and 97% at West of Shaggar) kept pet animals on the farm. The presence of pets and other animals can potentially mechanically transmit pathogens [30]. Surprisingly, 21.4%, 15.7% and 35% of the farms did not practice an all-in/all-out system. All these suggest a poor level of biosecurity practices by farmers and workers.

### 3.3. Assessment of External Biosecurity Practices

The assessment of external biosecurity practices across the total farms (*n*= 226) comprised eight subcategories, and the respective findings are shown below.

#### 3.3.1. Routines on Chicken Purchasing, Feeding and Watering Practices

The routines on chicken purchasing, feeding and watering practices of the poultry farms in the study areas are presented in Table 4.

In all the study sites, a significantly higher proportion of the poultry owners purchase day-old chicks (χ^2^ = 10.2; *p* = 0.006). Although the majority of poultry farmers obtained their chicken from reputable suppliers, there was no significant difference among the various sources of chicken suppliers (χ^2^ = 8.6; *p* = 0.196). For instance, in Bishoftu, the highest proportion of farms 46 (65.7%) purchased day-old chickens. In all three study sites, good biosecurity practices were noted in terms of obtaining chickens from well-known suppliers. There was not much batch mixing, and purchasing feed from well-known producing companies occurred. These good biosecurity practices were in alignment with the standard recommendations [15]. On the contrary, the majority of the farms performed no routine inspection while purchasing the chickens. There was no separate per farm delivery of purchased chickens. The results of the study revealed that 73.2%, 82.9% and 98% of the farms in Addis Ababa, Bishoftu and West of Shaggar, respectively, lack separate delivery of purchased chickens. Such variations among farms might arise from differences in farm managers’ knowledge and altitude about biosecurity practices related to chicken purchasing inspection and delivery practices [24,31]. The practices related to inspection and delivery of purchased chickens should be critically considered in view of the increased risk of disease introduction [32]. In addition, some of the farms did not have separate sealed feed storage to protect the feed from vermin and spilled water. Similarly, the practices of having sealed feed storage against water and vermin were found to be essential preventive methods to avoid feed contamination. Ensuring feed not to be contaminated with water and vermin averts the risk of pathogen transmission [9].

#### 3.3.2. Routines on Management of Wastes and Farm Entry Restriction Practices

The vast majority of the poultry farms in Addis Ababa (82.1%) and Bishoftu (77.1%) did not have separate waste disposal systems (Table 5). Meanwhile, 74% of the farmers at West of Shaggar had waste disposal systems. This difference in waste disposal systems among the study sites was statistically significant (χ^2^ = 64.7; *p* < 0.001). The results of this study revealed that a higher proportion of poultry farmers did not use personal protective equipment during waste handling and other operations. For instance, 69.6%, 65.7% and 69% of the farmers in Addis Ababa, Bishoftu and West of Shaggar, respectively, did not use gloves while removing waste materials. No statistically significant difference (*p* > 0.05) was observed among the study sites.

The observations of the present study dealing with waste management were in agreement with the study conducted in Gharbia Governorate, Egypt by [33], who similarly reported that about 75% of farms had no special designated area for poultry waste disposal and more than 85% never wore protective gloves or protective masks.

The farms included in this study disclosed the status of farm entry restriction practices. As presented in Table 5, 89.3%, 84.3% and 96% of the poultry farms did not register visitors coming to their farms, and only very few of the farms (1.8% in Addis Ababa, 20% at Bishoftu and 4% at West of Shaggar) provided farm-specific clothes. Surprisingly, 26.8%, 5.7% and 54% of the poultry farms in Addis Ababa, Bishoftu and West of Shaggar, respectively, indicated that their farm attendants/employees work on other poultry farms. Visitors can potentially act as vectors for transmission of pathogenic agents in the farms [34,35]. Human movements between farms are believed to have been a key factor in the spread of highly pathogenic avian influenza in the 2003 outbreak in The Netherlands [36,37]. Due to the absence of farm-specific protective clothes, employers working for different farms coupled with the absence of handwashing and disinfection during entry may favor the transmission of diseases [35,38].

#### 3.3.3. Routines on Material Sharing, Farm Infrastructure and Biological Factors 

The practice of sharing materials with other farms was quite evident and observed among 44.6%, 97.1% and 94% of the farms in Addis Ababa, Bishoftu and West of Shaggar, respectively (Table 6). Some of the farmers did not disinfect the materials shared before use. This was revealed by 37.5%, 27.1% and 85% of the farms in Addis Ababa, Bishoftu and West of Shaggar, respectively.

Aside from the role of shared materials, the absence of proper disinfection of visitors and vehicles can also serve as a disease transmission vector [39]. Most farms have protective wall materials made of brick wall and mesh wire, hindering access of chickens to open air and also access of pet animals to barns. As presented in Table 6, the access of wild birds to the poultry houses was found to be a common challenge observed in 60.7%, 64.3% and 18% of the farms of Addis Ababa, Bishoftu and West of Shaggar, respectively. Similarly, access of pet animals, mainly cats and dogs, to the farms was another challenging scenario. The high degree of manifestation of vermin (rats, mice, etc.) was noted in all three study sites. Vegetation cover with the potential to harbor animals and rodents is available on 91.1% of the farms in Addis Ababa and 32.9% of the farms in Bishoftu, whereas none of the farms at West of Shaggar had vegetation cover. Rodents were observed in 98.2%, 38.6% and 65% of the farms in Addis Ababa, Bishoftu and West of Shaggar, respectively. It was also indicated that bushy surroundings around poultry farms would allow breeding of insects and rodents [39]. Furthermore, pathogens can be introduced on the farm by rodents, wild birds and insects but also via pet animals and other farm animals serving as biological as well as mechanical vectors of pathogens [35]. It has been clearly demonstrated that rodents are an important vector of *Salmonella* Typhimurium and *Salmonella* Enteritidis, which are among the worldwide top five serotypes responsible for human infections [34,39,40]. For instance, wild birds are responsible for the spread and occurrence of various pathogens, mainly avian influenza virus, Newcastle Disease virus, *Mycoplasma* spp., *Campylobacter* spp., *Salmonella* spp., *Yersinia* spp. and *Mycobacterium avium* [34,35,41,42,43,44,45].

#### 3.3.4. Farms’ Relative Location

The present study disclosed that the majority of the poultry farms had close proximity to the main road that has access to various vehicles. For instance, about 55% and 42% of the poultry farms were situated within 100 m of the main road in Addis Ababa and West of Shaggar, respectively (Table 7). Only 3.6%, 18.6% and 35% of the farms were located farther than 200 m away from the main roads in Addis Ababa, Bishoftu and West of Shaggar, respectively. Similarly, a high number of farms were found close to human residence areas, within 100 meters (82.1% in Addis Ababa, 62.9% Bishoftu and 95% in West of Shaggar). Nearly 90 % of the farms are located less than 1 km from each other. Worth mentioning is that 66.1%, 34.3% and 69% of the farms in Addis Ababa, Bishoftu and West of Shaggar, respectively, were located less than 500 m from each other (Table 7). Chi-squared tests revealed a statistically significant association (*p* < 0.05) among the three variables (close proximity to main road, the proximity to human residence area and approximate distance from nearest poultry farm) and the poultry farms.

The relative farm location remains crucial because the close proximity of the farms to each other favors the increased likelihood of airborne pathogen transmission [23,46,47]. According to Geladude et al., 2014, a minimum distance of 500 me between two different poultry farms (preferably more than 1 km) may significantly reduce the risk of spread of airborne transmission of pathogens between poultry farms. This distance also applies to the location of a farm with respect to hobby poultry farms [35,47]. The present study disclosed that the relative placement of the farms from neighboring farms, human houses and the major roads was the first severely violated component of external biosecurity. This observation that revealed the compactness of poultry farms was comparable to a study conducted elsewhere [33]. On the other hand, the results of this study were quite different from those of studies conducted in developed countries such as Scotland on small- and medium-scale poultry farms. In the assessment in Scotland, most of the respondents (>50% overall) had seldom or never seen neighbors’ poultry and livestock farms within 100 meters [48].

### 3.4. Internal Biosecurity Assessment Practices

#### 3.4.1. Routines on Disease Management Practices

The routine disease management practices of the poultry farmers were assessed. The results revealed that 71.4%, 95.7% and 55% of the farms were practicing fixed vaccination programs in Addis Ababa, Bishoftu and West of Shaggar, respectively (Table 8). These results concur with the previous findings, which reported an encouraging status in disease diagnosis and vaccination programs (92.8%) in different parts of Ethiopia, including the Tigray, Amhara, Oromia and Addis Ababa regions [7]. However, the result is different from the report from Kenya since most farmers (38.5%) do not vaccinate their birds against the common preventable diseases [49]. Although poultry farmers practiced chicken health monitoring in all the study sites, the frequency of monitoring was variable. The frequency of practice of health status monitoring was extended, being more than 2 weeks in Bishoftu (47.1%), West of Shaggar (86%) and Addis Ababa (71.4%), where they often monitored flock health status every week or less. About 28.1% of the farms did not achieve professional health monitory services. On the other hand, 53.6%, 27.1% and 59% of the farms found in Addis Ababa, Bishoftu and West of Shaggar had no separate rooms and practices for the removal of dead birds (Table 8).

Among disease management practices, vaccination remains as the main prophylactic measure to combat highly pathogenic diseases of poultry along with other activities, such as removing dead birds [46,50,51].

#### 3.4.2. Routines on Cleaning and Disinfection Practices

Observation was also carried out on the cleaning and disinfection practices of the poultry farms. Cleaning of the premises was well-practiced after each production cycle on the majority of the farms in Addis Ababa (91.1%) and Bishoftu (77.1%), whereas it was less practiced by poultry farms at West of Shaggar (56%) (Table 9).

Surprisingly, the absence of a footbath was observed on 58.9% of the farms in Addis Ababa and 71% of the farms in West of Shaggar. Encouragingly, 100% of the poultry farms had the presence of a footbath facility in Bishoftu. Although a footbath was available, visitors and workers had access to the interior of the farms without using the footbath in 58.9% in Addis Ababa and 71% in West of Shaggar, while it was the case in only 2.9% of the farms in Bishoftu (Table 9). With regard to the level of practice of cleaning and disinfection of farm materials, inconsistency in its applicability was observed.

Cleaning and disinfecting are of great importance for the control of diseases in poultry. It should be avoided for chicks to come into contact with litter, dust, feathers and other debris from the previous flock [52,53]. Some pathogens can survive for a long time in the environment without the presence of poultry [54]. Therefore, the following steps of the complete cleaning and disinfection protocol should be carried out between two production cycles: dry cleaning, wet cleaning, disinfection, vacancy period and monitoring the efficacy [55,56]. Not only the interior of the stables (including the drinking and feeding lines) but also the environment around the stables may form a potential reservoir for several pathogens, including *Campylobacter* spp. [57]. *Campylobacter* spp. is a significant cause of bacterial zoonosis responsible for enterocolitis in humans and contamination of poultry flocks at the farm level, and it often leads to transmission of *Campylobacter* along the poultry production chain and contamination of poultry meat at retail [58]. The findings of the later study also show the importance of taking hygienic measures before entering the poultry house.

### 3.5. Adoption Level of Biosecurity Components

The level and extent of the external, internal and overall biosecurity practices with regard to the ten components and the comparison with the global average are indicated in Table 10. Concerning external biosecurity practices, it was noted that feed and water management of the poultry farms were highly implemented across most farms at 69.9%, even higher than the global average, which is 58%. On the contrary, farm relative location, farm infrastructure status, inter-farm material sharing, farm entry restrictions and waste and manure management practices were vastly violated by most of the farms, resulting in the low level of implementation with a statistically significant variation (*p* < 0.05) observed as opposed to the respective global average. Within the category of external biosecurity, chicken purchasing practices and feed water management practices showed no statistically significant variation (*p* > 0.05) with the respective global average [15,24]. In general, the external biosecurity score was significantly lower than the global average (40.7 versus 64.6; *p* < 0.001). From the internal biosecurity point of view that encompasses disease management, cleaning and disinfection practices in central Ethiopia were implemented by 52.6% as opposed to the global average of 64%. Similarly, the levels of those two internal biosecurity components were also below the global average (DM—disease management 48.98% vs 73% and CD—cleaning and disinfection 56.2% vs 61%) (Table 10). 

However, there was no statistically significant difference (*p* > 0.05) between the scores of disease management and cleaning and disinfection practices and those of the global average [15,24]. The overall, external and internal biosecurity scores of the studied poultry farms in the three study areas along with their subcomponents were demonstrated in Table 10. The overall biosecurity practices for the three study sites were computed by considering 80% for external and 20% for internal scores. The findings of the present study revealed no observed uniformity among farms in their compliance with most of the subcategories of external and internal biosecurity scores. As a result, the biosecurity scores for external, internal and overall were noted to be lower than the global average. The global averages for the overall, external and internal biosecurity practices were obtained from Biocheck.Ugent, (Ghent University, Merelbeke, Belgium) [15].

It was disclosed that the overall biosecurity score of 43.1 for central Ethiopia was far below the global average score of 64.3 [24]. The external and internal biosecurity scores prior to conversion to 80% and 20% were 40.7 and 52.6, respectively (Table 10). A comparison of the findings of the present study to the highly developed European countries indicated wider gaps in the mean overall biosecurity score (43.1 vs 70.9). Similarly, the observed mean external (40.7) and internal (52.6) biosecurity scores were much lower than the European average (68 external and 76.6 internal) [23].

The boxplots for overall biosecurity practices for the three study sites revealed differences as depicted in Figure 2. Accordingly, the median biosecurity score was highest for farms in Bishoftu and lowest for farms in West of Shaggar.

The variations in the overall biosecurity practices in the three study sites could be linked to the type of farms and their management systems. In Bishoftu, all the commercial large-scale farms were mainly concentrated, and extensive farming experience might contribute to the highest overall biosecurity level compared to small- and medium-scale farms [27].

## 4. Conclusions

In a nutshell, the current study on the assessment of biosecurity implementations for small-, medium- and large-scale poultry farms revealed a lack of application of various external and internal biosecurity components and indicated room for improvement. Principally, six of the eight components of the external biosecurity measures were highly disrupted. Likewise, internal biosecurity practices were implemented lower than the global average, indicating lower performance and relaxation in biosecurity measures. Accordingly, biosecurity practices comprising external and internal components with low scores must be ranked for further emphasis and improvement. Considering the above-concluding remarks, the following recommendations are forwarded. The government at large and the Ministry of Agriculture in particular should be actively involved in supporting and participating in enhancing biosecurity practices to ultimately halt the existing widespread occurrence of diseases and spread at the national level. The agricultural sector of Ethiopia should place emphasis on the provision of extension services to overcome the low level of biosecurity practices in the poultry farms of the study area and beyond. Hygiene and sanitation play a major role in any effective disease control program for poultry production premises. Thus, it is critically important to prioritize biosecurity guidelines and strategies, awareness creation campaigns, advocacy on the concept and implementation of biosecurity and pertinent contextual training to all relevant sectors and professionals. The enforcement of the application of biosecurity practices should be an obligatory prerequisite for license renewal and establishment of new farms.

## Figures and Tables

**Figure 1 animals-13-03719-f001:**
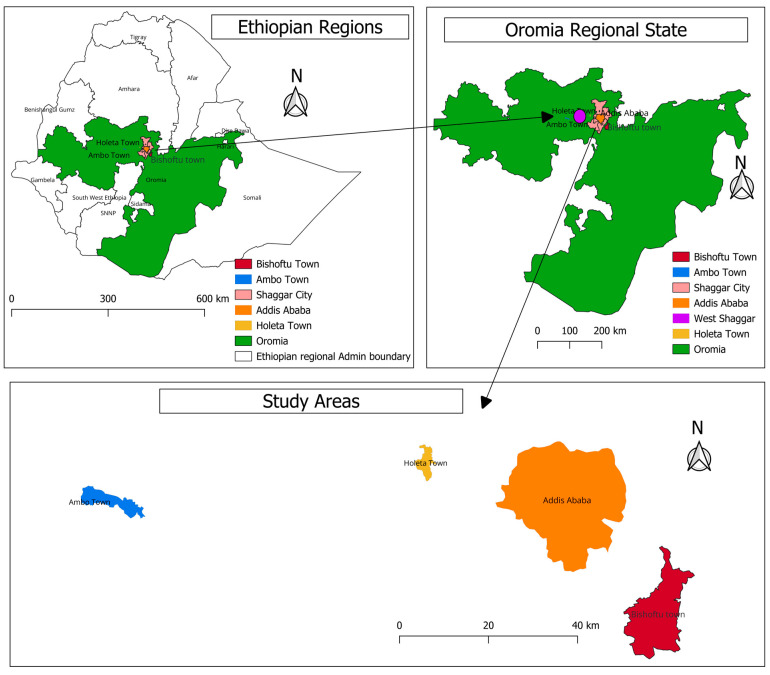
Map of the study areas where samples were collected using QGIS Ver. 3.14 (QGIS Development Team, 2009. QGIS Geographic Information System. Open Source Geospatial Foundation. http://qgis.org (accessed on 27 November 2023)).

**Figure 2 animals-13-03719-f002:**
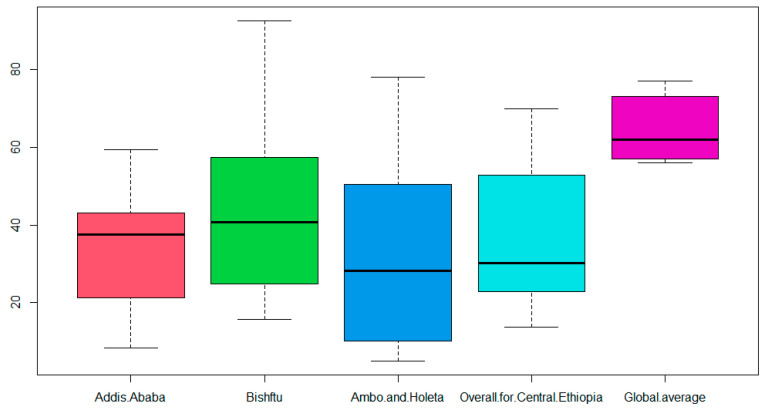
Boxplots of overall biosecurity scores by study sites.

**Table 1 animals-13-03719-t001:** Operational definitions of the external and internal biosecurity components.

External Biosecurity Components	Definition
Chicken Purchasing Practice (CPP)	Refers to type of chicken purchased, source of purchased chicken, delivery of chicken from same source, inspection routine while purchasing and separate delivery of purchased chicken
Feed and Water Management (FWM)	Indicates practices that assess the source of feed, sealed feed storage against water and vermin and source of drinking water
Waste and Manure Management (WMM)	Denotes waste and manure management practices comprising separated waste disposal area, way of handling wastes, use of gloves during waste handling, habit of handwashing after waste handling and destination of farm wastes
Farm Entry Restriction (FER)	Shows practices consist of obligation of visitors to register, presence of farm specific clothes, handwashing and disinfection during farm entry, employees working in different farms
Inter-farm Material Sharing (IMS)	Encompasses inter-farm material sharing practices and disinfect shared materials when receiving to use
Farm Infrastructure Status (FIS)	States farm infrastructure status mainly the material used to construct the wall of the chicken farm
Control of Biological Vector (CBV)	Include the situation of control of biological vector practices that encompasses access of birds to outside (open air), vegetation potentially harbors other animals, manifestation of vermin (e.g., rats, mice, etc.) and access of pet animals (cats and dogs)
Farm Relative Location (FRL)	Denotes the position of farm relative location in terms of farm relative location from main road, residence area close to farm location and the approximate distance from nearest poultry farm
**Internal Biosecurity Components**	**Definition**
Disease Management (DM)	Deals with disease management practices including the extent and degree of fixed vaccination program, monitory of health status, professional help for health status monitory, isolation of sick birds and removal of dead birds
Cleaning and Disinfection (CD)	Refers to cleaning and disinfection practices that include biosecurity practices pertaining to cleaning of poultry farm after each production cycle, presence of footbath facility, probability of accessing farm without footbath, frequency of changing footbath and cleaning and disinfection of farm materials.

**Table 2 animals-13-03719-t002:** The demographic characteristics of study participants in selected areas of central Ethiopia (*n* = 226).

Variables	Categories	Addis Ababa N = 56	Bishoftu N = 70	West of Shaggar N = 100
Role of respondent	Farm owner/co-owner	82.1%	52.9%	88%
	Employee	16.1%	41.4%	8%
	Owner’s relative	1.8%	5.7%	4%
Sex	Male	60.7%	62.9%	58%
	Female	39.3%	37.1%	42%
Age	18–25 years	1.8%	20%	15%
	26–35 years	51.8%	52.9%	49%
	36–45 years	35.7%	22.9%	27%
	>45 years	10.7%	4.3%	9%
Educational Status	No formal education	7.1%	7.1%	9%
	Primary school	26.8%	35.7%	28%
	Secondary school	42.9%	28.6%	26%
	Diploma and above	23.2%	28.6%	37%

N: Number of respondents.

**Table 3 animals-13-03719-t003:** Characteristics of the studied chicken farms in selected sites of central Ethiopia (*n* = 226).

Biosecurity Measures	Responses	Addis Ababa N = 56	Bishoftu N = 70	West of Shaggar N = 100	*Chi-square*	*p*-Value
Purpose of chicken production	Egg	85.7%	64.3%	64%	40.2	<0.001
	Meat	14.3%	30%	8%		
	Dual	0%	5.7%	28%		
Keeping different types of breeds	Yes *	14.3%	2.9%	7%	5.9	0.052
	No	85.7%	97.1%	93%		
Age of chickens	<2 months	14.3%	30%	14%	17.1	0.009
	2–6 months	14.3%	25.7%	29%		
	7–12 months	42.8%	37.1%	35%		
	>12 months	28.6%	7.1%	22%		
Housing system	Deep litter system	83.9%	78.6%	99%	19.9	0.001
	Cage system	12.5%	18.6%	1%		
	Both	3.6%	2.9%	0%		
Breeds of chickens	Bovans Brown	85.7%	47.1%	57%	56.7	<0.001
	Sasso	0%	5.7%	28%		
	Cobb 500	14.3%	30%	8%		
	Lohmann Brown	0%	17.1%	7%		
Size of farm	Small scale	73.2%	47.1%	79%	51.6	<0.001
	Medium scale	26.8%	24.3%	21%		
	Large scale	0%	28.6%	0%		
All-in/all-out practice	Yes *	78.6%	84.3%	65%	8.7	0.013
	No	21.4%	15.7%	35%		
Presence of pet animals	Yes	100%	41.4%	97%	99.1	<0.001
	No	0%	58.6%	3%		

* Number and percentage of respondents keeping different types of breed separately “Yes”; N: number of studied chicken farms.

**Table 4 animals-13-03719-t004:** Routines on chicken purchasing practice, feeding and watering (*n* = 226).

Biosecurity Measures	Responses	Addis Ababa N = 56	Bishoftu N = 70	West of Shaggar N = 100	*Chi-square*	*p*-Value
Type of chicken purchased	Day-old chicken	53.6	65.7%	41%	10.2	0.006
	Pullets	46.4%	34.3%	59%		
Source of purchased chicken	Well-known supplier	78.6%	85.7%	82%	8.6	0.196
	Local chicken supplier	14.3%	14.3%	10%		
	Middleman	7.1%	0%	8%		
Delivery of chicken from same source	Yes, always the same source	10.7%	72.9%	55%	89.3	<0.001
	No, sometimes vary	46.4%	25.7%	43%		
	No, mostly vary	42.9%	1.4%	2%		
Batch mixing practices	Yes	21.4	15.7%	35%	8.7	0.013
	No	78.6%	84.3%	65%		
Inspection routine while purchasing	Overall examination	17.9%	15.7%	18%	23.3	<0.001
	Random size and/or weight	12.5%	34.3%	7%		
	No known inspection	69.6%	50%	75%		
Separate delivery of purchased chicken	Yes *	26.8%	17.1%	2%	21.4	<0.001
	No	73.2%	82.9%	98%		
Source of feed	Purchased from companies	69.6%	94.3%	95%	42.7	<0.001
	In-house manual feed mix	7.1%	5.7%	55%		
	Mixed	23.2%	0%	0%		
Sealed feed storage against water	Yes *	71.4%	98.6%	77%	19.2	<0.001
	No	28.6%	1.4%	23%		
Sealed feed storage against vermin	Yes *	8.9%	82.9%	58%	69.8	<0.001
	No	91.1%	17.1%	42%		
Source of drinking water	Tap water	87.5%	94.3%	1%	221.2	<0.001
	Well water	0%	0%	98%		
	Mixed	7 (12.5%)	5.7%	1%		

* Number and percentage responded “Yes”; N: Number and percentage of respondents per study site. N reference to number of Respondents per study site.

**Table 5 animals-13-03719-t005:** Routines on management of waste and farm entry restriction practices (*n* = 226).

Biosecurity Measures	Responses	Addis Ababa N = 56	Bishoftu N = 70	West of Shaggar N = 100	*Chi-square*	*p*-Value
Separated waste disposal area	Yes *	17.9%	22.9%	74%	64.7	<0.001
	No	82.1%	77.1%	26%		
Handling waste	Composting in pit	16.1%	54.3%	67%	69.3	<0.001
	Stored in sealed bag	25%	15.7%	15%		
	Immediate removal	17.9%	24.3%	16%		
	No recognized system	41.1%	5.7%	2%		
Use of gloves during waste handling	Always	1.8%	2.9%	6%	6.8	0.555
	Sometimes	28.6%	31.4%	25%		
	Never	69.6%	65.7%	69%		
Washing after waste handling	Always	48.2%	80%	70%	17.2	0.002
	Sometimes	50%	18.6%	26%		
	Never	1.8%	1.4%	4%		
Destination of farm waste	Disposed around farm	28.6%	62.9%	12%	155.8	<0.001
	Sell for other uses	5.4%	25.7%	51%		
	Taken by dirt collectors	12.5%	1.4%	37%		
	Mixed	53.6%	10%	0%		
Visitors register	Yes *	10.7%	15.7%	4%	6.9	0.032
	No	89.3%	84.3%	96%		
Presence of farm specific clothes	Yes, always	1.8%	20%	4%	55	<0.001
	Yes, sometimes	33.9%	41.4%	7%		
	No	64.3%	38.6%	89%		
Handwashing and disinfection during farm entry	Always	10.7%	35.7%	4%	95.6	<0.001
	Sometimes	53.6%	44.3%	9%		
	Never	35.7%	20%	87%		
Employees working in different farms	Yes *	26.8%	5.7%	54%	44.9	<0.001
	No	73.2%	94.3%	46%		

* Number and percentage responded per farm “Yes”; N: Number and percentage of respondents per study site.

**Table 6 animals-13-03719-t006:** Routines on material sharing, farm infrastructure and biological factors (*n* = 226).

Biosecurity Measures	Responses	Addis Ababa N = 56	Bishoftu N = 70	West of Shaggar N = 100	*Chi-square*	*p*-Value
Material being shared with other farms	Always	6 (10.71%)	0%	1%	96.6	<0.001
	Sometimes	19 (33.93%)	97.1%	93%		
	Never	55.4%	2.9%	6%		
Disinfect materials after receiving prior usage	Always	16.1%	28.6%	4%	68.1	<0.001
	Sometimes	46.4%	44.3%	11%		
	Never	37.5%	27.1%	85%		
Material of chicken farm wall made of	Brick wall	42.9%	40%	6%	57	<0.001
	Wire mesh	51.8%	34.3%	53%		
	Soil plastered wall	3.6%	10%	31%		
	Bamboo and others	1.79%	15.7%	10%		
Access of chickens to the outside (open air)	Yes	26.9%	5.7%	23%	11.4	0.003
	No *	73.2%	94.3%	77%		
Access of wild birds to the farm	Yes	60.7%	64.3%	18%	45.6	<0.001
	No	39.3%	35.7%	82%		
Vegetation potentially harbors other animals	Yes	91.1%	32.9%	0%	135.2	<0.001
	No*	9%	67.1%	100%		
Manifestation of vermin (e.g., rats, mice, etc.)	Yes	98.2%	38.6%	65%	48.7	<0.001
	No *	1.8%	61.4%	35%		
Access of pet animals (cats and dogs)	Yes	26.8%	42.9%	4%	37.8	<0.001
	No *	73.2%	57.4	96%		

* Number and percentage responded “No”; N: Number and percentage of respondents per study site.

**Table 7 animals-13-03719-t007:** Geographical locations of the poultry farms relative to main road, residence area and other poultry farms (*n* = 226).

Biosecurity Measures	Responses	Addis Ababa N = 56	Bishoftu N = 50	West of Shaggar N = 100	*Chi-square*	*p*-Value
Farm relative location from main road	Main road within 100 m	55.4%	34.3%	42%	27.5	<0.001
	Main road 100–200 m	41.1%	47.1%	23%		
	Main road > 200 m	3.6%	18.6%	35%		
Residence area close to farm location	House within 100 m	82.1%	62.9%	95%	29.6	<0.001
	House within 100 –200 m	8.9%	24.3%	3%		
	House > 200 m	8.9%	12.9%	2%		
Approximate distance from nearest poultry farm	<500 m	66.1%	34.3%	69%	30.1	<0.001
	Between 500 m and 1 km	21.4%	41.4%	25%		
	>1 km	12.5%	24.3%	6%		

N: Number and percentage of respondents per study site.

**Table 8 animals-13-03719-t008:** Responses on disease management practices (*n* = 226).

Biosecurity Measures	Responses	Addis Ababa N = 56	Bishoftu N = 50	West of Shaggar N = 100	*Chi-square*	*p*-Value
Fixed vaccination program	Always followed	71.4%	95.7%	53%	36.4	<0.001
	Sometimes followed	28.6%	4.3%	47%		
Monitory of health status	Every week or less	71.4%	12.9%	2%	135.5	<0.001
	Every two weeks	19.6%	40%	12%		
	More than two weeks	8.9%	47.1%	86%		
Professional help for health status monitory	Private animal health workers	16.1%	84%	58%	186.3	<0.001
	Government animal health worker	8.9%	4.3%	40%		
	By farm workers	75%	7.1%	2%		
Keeping different age groups	Yes	78.6%	10%	8%	6.5	0.038
	No	21.4%	90%	92%		
Removal of dead birds	Separated in isolation room	8.9%	27.1%	59%	42	<0.001
	Separated at corner of room with apparently healthy flocks	37.5%	64.3%	17%		
	No separation room	53.6%	27.1%	59%		
Isolation of sick birds	Immediately after observation	5.4%	67.1%	98%	81.1	<0.001
	Can be kept up to 24 h	28.6%	32.9%	2%		
	Can be kept for > 24 h	5.4%	0%	0%		

N: Number and percentage of respondents per study site.

**Table 9 animals-13-03719-t009:** Responses on cleaning and disinfection practices (*n* = 226).

Biosecurity Measures	Responses	Addis Ababa N = 56	Bishoftu N = 50	West of Shaggar N = 100	*Chi-square*	*p*-Value
Cleaning of poultry farm after each production cycle	Always	91.1%	77.1%	56%	71.3	<0.001
	Sometimes	8.9%	22.7%	44%		
Presence of footbath facility	Yes *	41.1%	100%	29%	88.6	<0.001
	No	58.9%	0%	71%		
Probability of accessing farm without footbath	Never	41.1%	81.4%	29%	91.8	<0.001
	Sometimes	0%	15.7%	0%		
	Frequently	58.9%	2.9%	71%		
Frequency of changing footbath	Everyday	1.8%	2.9%	1%	100.7	<0.001
	Every two days	12.5%	55.7%	5%		
	Every three or more days	26.8%	40%	23%		
	Not applicable	58.9%	1.4%	71%		
Cleaning and disinfection of farm materials	Always	91.1%	77.1%	56%	23.3	<0.001
	Sometimes	8.9%	22.9%	44%		

* Number and percentage responded “Yes”; N: Number and percentage of respondents per study site.

**Table 10 animals-13-03719-t010:** The status of external, internal and overall biosecurity scores of the poultry farms (*n* = 226) in the study areas, central Ethiopia in comparison with the global average.

External Biosecurity Components	Addis Ababa (%)	Bishoftu (%)	West of Shaggar (%)	Overall for Central Ethiopia (%)	Global Average (%)	*t*-Test, *p* Value
CPP	43.2	51.4	41.8	45.5	56	0.1440
FWM	59.4	92.5	57.6	69.9	58	0.2347
WMM	18.2	31.1	43	30.8	59	0.0012
FER	24.1	41.4	14.5	26.7	69	0.0005
IMS	35.7	15.7	5	18.8	56	0.0066
FIS	42.9	40	6	29.6	77	0.0571
CBV	39.3	63.1	78	60.1	77	0.0494
FRL	8.3	18.6	14.3	13.8	65	0.0000
Overall external biosecurity				40.7	64.6	0.0000
Overall external biosecurity computed out of 80%				32.6		
DM	34.6	58.3	54	49	73	0.0163
CD	55.4	78.3	35	56.2	61	0.5423
Overall internal biosecurity				52.6	64	0.0578
Overall internal biosecurity computed out of 20%				10.5		
Overall biosecurity practices *				43.1		
Overall biosecurity global average				64.3		

* Total was obtained after converting external and internal biosecurity values to 80% and 20%, respectively, and computing the average. Abbreviations: CPP (Chicken Purchasing Practice), FWM (Feed and Water Management), WMM (Waste and Manure Management), FER (Farm Entry Restriction), IMS (Inter-farm Material Sharing), FIS (Farm Infrastructure Status), CBV (Control of Biological Vector), FRL (Farm Relative Location), DM (Disease Management) and CD (Cleaning and Disinfection).

## Data Availability

The data in the present study are available from the corresponding author upon reasonable request.

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
