# Peer review of "Quantitative Assessment of Major Biosecurity Challenges of Poultry Production in Central Ethiopia"

_animals, 2023, doi:10.3390/ani13233719_

Round 1

Reviewer 1 Report

Comments and Suggestions for Authors

Minor Comments

1. Line 168; "The results revealed that the purpose of chicken farming was predominately layer." This does not sound correct; you can write that the results revealed that the purpose of chicken farming was predominately for egg production. Please correct similarly throughout the manuscript. 

2. Line 169; stock? Flock? 

3. Line 229; please correct the in-text citation. 

4. There are many typos throughout the manuscript. Please read carefully and correct those. 

Major Comments

1. "This study investigated the biosecurity status of 82 poultry farms in central Ethiopia where a large number of commercial farms of various 83 scales (large, medium and small) were investigated." Line 82-84; this is not necessary in the Introduction. 

2. Line 94-95. What was the basis of the classification of farms in three categories? In lines 105-106, I saw the number of birds was the basis, but why did you choose this classification? Did you use any references? If yes, please cite that. For example, 5,000 birds in a farm might be less in another country, so if you cite some papers, the reader will have a better idea. 

3. In 2.2, study population, write some details about the farm statistics in the study area. 

4. Line 110, "Afterwards, the biosecurity audit was performed during an unannounced farm visit." Why did you choose to visit without informing the farmers? To whom do you collect data? Famers? Workers? Since this is survey research, there are high chances of bias. If you take data from farmers vs workers, there might be differences. 

5. In Table 3, in Addis Ababa, I saw Age of chickens, with less than 2 months, is 0%. Similarly, in the same column, Broiler is 14.3%. All 8 farms had broilers that were more than 60 days old. What is the market age of broilers in Ethiopia? 

6. In the same table, in Bishoftu, 30% are broilers. But I could see all layers breed; none are broiler breed. How do you justify this? Also, what is double in "Purpose of chicken production"? Is it the same breed with a dual-purpose or mixed-rearing system, or do farmers have both broilers and layers? Write it clearly. 

7. Line 186: "All these suggest the poor awareness of the owners on the biosecurity scenario." This is not poor awareness; this is poor practice. There is a difference in awareness and practice; farmers can be aware but may not practice due to various reasons. As a researcher, you must give careful attention to such details in the survey study. 

8. How did you define "reputable suppliers"?

9. You have mentioned Salmonella Typhimurium and Salmonella Enteritidis in line 268. Why did you put it there? Why not Salmonella Gallinarum and Pullorum. Please mention the food safety role of Typhimurium and Enteritidis. 

10. What is the role of Campylobacter spp. in poultry health? Please mention in line 344. 

11. I suggest writing limitations, future research direction, and limitations of the study at the end of the discussion. 

12. Line 405-407. "The government at large and the ministry of agriculture, in particular, should actively involve and participate in enhancing biosecurity practices to ultimately halt the existing widespread occurrence of diseases and spread at the national level." Is this the conclusion of your study? Please rewrite the exact conclusion of your study and move the rest of the things into discussion. 

Comments on the Quality of English Language

Though I am not a native English speaker, I suggest moderate editing of the English language. 

Author Response

A point-by-point response to the valuable comments raised by reviewer number One (I) is kindly attached  and please see the attachment. 

Reviewer 2 Report

Comments and Suggestions for Authors

This is a well-conceived and well-written study. The low level of biosecurity found in the study is somewhat surprising given what we now know about the importance of biosecurity in disease prevention, particularly in the case of avian influenza. The large number of farms that do not practice all in-all out production is somewhat concerning from a disease potential standpoint (Table 3), as is the lack of sealed feed storage (Table 4). Management routines on wastes, farm entry restrictions, material sharing, and biological are also concerning but not overly shocking.  However, it does bring into question the level of support farmers and company managers may be receiving from Ethiopian agricultural extension services.  Your assessment certainly highlights the biosecurity challenges associated with poultry production. You highlight the importance of isolation, traffic control and cleaning/sanitation/disinfection but why is this not being done?  Is it a lack of training and knowledge of biosecurity principles or are principles taught and known but not being followed?  If you have knowledge of assistance (or lack of) from local agricultural extension officers, it would be useful to include that knowledge. Knowing that biosecurity practices are not being followed is important but perhaps just as important is why those practices are not being followed. 

Did you gain any insight as to how the individuals became aware of biosecurity practices and if there was any on-farm biosecurity training programs?  Based on information in Table 5, it seems training make be lacking on the farm considering the lack of visitors logs and the number of employees that work on multiple farms particularly in Addis Ababa and West Sheger.

The amount of material being shared between farms (Table 6) also is concerning from a biosecurity standpoint.  Were you aware if this equipment was cleaned first and then disinfected before it left the farm of origin and again before it returned? You can't disinfect organic matter.  If dirty materials were disinfected but not cleaned first, how much good was disinfecting doing?

Lack of vegetation control and no rodent control program hints at no on-farm biosecurity training programs in place.  However, it would be useful if you could comment on this or whether anyone anywhere is offering biosecurity training.  The difference in the use of footbaths at the three locations does seem to indicate that at least some form of training is taking place at some locations.  It would be useful to know if this is at the farm level or through extension services or how it is being provided.

The overall low level of biosecurity in use certainly indicates more must be done but knowing where to start will hinge somewhat on where the gaps currently exist. If you have insight into that, it would help to strengthen your paper.

A very useful paper that provides much insight into the gaps in poultry biosecurity in Ethiopia and helps explain the ongoing disease threat.

Author Response

A point-by-point response to the valuable comments raised by reviewer number Two (II) is kindly attached  and please see the attachment. 

Round 2

Reviewer 1 Report

Comments and Suggestions for Authors

Comments

1. Line 407, there was no statistically significant difference (P<0.05), I believe the arrow should be in the opposite direction if it is not significant. Please check this throughout the manuscript. 

2. I hope data, this time, are accurate. 

Comments on the Quality of English Language

Moderate editing of English language required

Author Response

Dear Editors and Reviewers,

Once again, thank you very much for your follow and effective communications in the processing of the manuscript designated as Animals 2731348 for publication in MDPI Animals Journal.

Following the second round of valuable comments received from only one of the reviewers, the authors went through and addressed the minor comments given and thus updated the manuscript accordingly. More specifically, one-by-one responses are given to the valuable comment raised in a separate document. The changes made in the manuscript are also indicated in track changes as well as highlighted in bright green. This time also authors reviewed minor typo errors and English grammar.

I would like to take this opportunity to duly acknowledge and appreciate the reviewers and your good office for the prompt review process.

This is therefore, to kindly submit to your good office the second round responses given to one of the reviewers(attached with this cover letter separately) and updated manuscript for your consideration and further process.

Sincerely,
